# Bromoperoxidase Producing *Bacillus* spp. Isolated from the Hypobranchial Glands of A Muricid Mollusc Are Capable of Tyrian Purple Precursor Biogenesis

**DOI:** 10.3390/md17050264

**Published:** 2019-05-03

**Authors:** Ajit Kumar Ngangbam, Peter Mouatt, Joshua Smith, Daniel L. E. Waters, Kirsten Benkendorff

**Affiliations:** 1Marine Ecology Research Centre, School of Environment, Science and Engineering, Southern Cross University, Lismore, NSW 2480, Australia; a.ngangbam.10@student.scu.edu.au; 2Southern Cross Plant Science, Southern Cross University, Lismore, NSW 2480, Australia; peter.mouatt@scu.edu.au (P.M.); joshua.smith@scu.edu.au (J.S.); dawaters@csu.edu.au (D.L.E.W.); 3ARC Industrial Transformation Training Centre for Functional Grains, Charles Sturt University, Wagga Wagga, NSW 2650, Australia

**Keywords:** shellfish purple, bacillus, bromoperoxidase, tyrindoxyl sulphate, whelk

## Abstract

The secondary metabolite Tyrian purple, also known as shellfish purple and royal purple, is a dye with historical importance for humans. The biosynthetic origin of Tyrian purple in Muricidae molluscs is not currently known. A possible role for symbiotic bacteria in the production of tyrindoxyl sulphate, the precursor to Tyrian purple stored in the Australian species, *Dicathais orbita*, has been proposed. This study aimed to culture bacterial symbionts from the purple producing hypobranchial gland, and screen the isolates for bromoperoxidase genes using molecular methods. The ability of bromoperoxidase positive isolates to produce the brominated indole precursor to Tyrian purple was then established by extraction of the culture, and analysis by liquid chromatography–mass spectrometry (LC–MS). In total, 32 bacterial isolates were cultured from *D. orbita* hypobranchial glands, using marine agar, marine agar with hypobranchial gland aqueous extracts, blood agar, thiosulphate citrate bile salts sucrose agar, and cetrimide agar at pH 7.2. These included 26 *Vibrio* spp., two *Bacillus* spp., one *Phaeobacter* sp., one *Shewanella* sp., one *Halobacillus* sp. and one *Pseudoalteromonas* sp. The two *Bacillus* species were the only isolates found to have coding sequences for bromoperoxidase enzymes. LC–MS analysis of the supernatant and cell pellets from the bromoperoxidase producing *Bacillus* spp. cultured in tryptone broth, supplemented with KBr, confirmed their ability to produce the brominated precursor to Tyrian purple, tyrindoxyl sulphate. This study supports a potential role for symbiotic *Bacillus* spp. in the biosynthesis of Tyrian purple.

## 1. Introduction

Many marine invertebrates produce secondary metabolites that contribute to a suite of ecological roles, including paralysing their prey and preventing predation [1], pathogens [2], surface fouling [3] and competitors [4]. Beyond the ecological roles of the secondary metabolites, these chemicals also provide opportunities that aid human society in the form of novel bio-products. A large number of marine natural products have been isolated and characterized from marine invertebrates [5], including antimicrobial, antifungal, antiviral, antiprotozoal, anthelminthic and anticancer compounds [6,7,8], as well as dyes and pigments [9,10,11,12].

The Muricidae are predatory gastropods of importance for their use in medicine and perfumery [13,14], as well as for the production of historically and culturally important purple and blue dyes [15,16,17]. The blue dye has been identified as indigo, a natural product well-known to be produced by plants [18,19] and bacteria [20,21,22], in addition to muricid molluscs. Tyrian purple is the dibrominated derivative of indigo (6,6′dibromoindigo), and along with the red structural isomer, 6,6′dibromoindirubin, it is only conclusively known to be produced by Muricidae molluscs [23]. Tyrian purple is currently regarded as a “rare” natural dye, with 6,6′dibromoindigo from shellfish currently valued at $4280 per gram [24]. The oxidised precursor 6-bromoisatin and a derivative of 6-dibromoindirubin are also of interest for their anticancer and anti-inflammatory properties [25,26,27,28,29,30,31].

Different species of Muricidae produce and store a range of brominated and nonbrominated indoxyl sulphate precursors, which form the purple and blue dyes respectively [32,33]. However, the Australian species *Dicathais orbita* produces a single brominated precursor tyrindoxyl sulphate (Figure 1a) and, therefore, has been identified as a useful model species for Tyrian purple production [23]. A transcriptomic study on *D. orbita* has identified a tryptophanase enzyme that can convert tryptophan to indoxyl sulphate [34]. However, the biosynthesis of tyrindoxyl sulphate in the hypobranchial gland of muricids, also requires bromoperoxidase to brominate the indole precursors [35,36,37]. The brominated indoxyl sulphate precursors are hydrolysed by an aryl sulfatase enzyme, which is produced by the mollusc [34], and then spontaneously reacts with oxygen, dimerises and is photolytically cleaved to form the final dye pigment [17,23].

There is a growing research interest in the involvement of bacteria–host invertebrate associations in the biosynthesis of marine secondary metabolites [38]. Structural homology between marine invertebrate natural products and microbial metabolites can provide an indication that these natural products originate from the microbes, rather than being synthesized by the host [39]. The structural similarity between Tyrian purple and indigo, coupled with the culture-independent identification of bacteria capable of producing indole and brominating secondary metabolites, in the *D. orbita* hypobranchial gland [40,41], suggests a potential role for bacterial symbionts in the biosynthesis of the precursor of Tyrian purple. Bacterial species, such as *Bacillus* sp., *Synechococcus* sp., and *Pseudomonas putida,* which were identified in *D. orbita* tissue [40], have been previously found to produce bromoperoxidase enzymes. Several other bacterial species are also known to produce bromoperoxidase [42,43,44,45,46] and indoles [47,48,49]

A major issue for the commercial development of marine secondary metabolites is sustainable supply [5,38,50]. Identification of bacteria involved in the biosynthesis of marine secondary metabolites can provide options for supplying sufficient quantities for clinical testing and commercialization [50,51,52,53]. Many marine bacteria cannot be easily cultured [54,55,56], however when possible, the ability to culture bacteria using traditional techniques provides an advantage for the large-scale production of natural products. Marine microbial symbionts associated with the host invertebrate might provide an alternative for producing compounds of biomedical importance, on a larger scale, for drug development [57,58,59,60] or dye production and could possibly solve the sustainable supply issue.

Many approaches have been used to “culture the unculturable” bacteria from marine invertebrates. As an example, the optimization of growth media, incubating temperature and pH increases the recovery of microbes that are unculturable under standard conditions [61,62,63]. Sterile marine water can be used to mimic the natural environment for culturing previously uncultured bacteria [61], while marine invertebrate extracts can also mimic the chemical environment within hosts [64,65]. For example, Li and Liu [66] used sponge extracts in artificial seawater to “simulate” the sponge natural environment, resulting in the isolation of bacteria belonging to *Actinobacterium* and *Bacteroidetes.*

The muricid, *Dicathais orbita,* is an ideal model species for studying the biosynthesis of Tyrian purple and bioactive brominated indole derivatives [13,23]. Some indole producing bacteria have been previously isolated from the hypobranchial gland of *D. orbita* [41], but their ability to produce brominated metabolites has not been determined. Recent culture-independent metagenomic and transcriptomic studies on the hypobranchial gland, have reported a large number of microbial symbionts and genes, some of which have the capacity of producing both indoles and brominated compounds [34,40]. This study aimed to culture bacteria potentially involved in Tyrian purple precursor synthesis, and then screen the isolates for bromoperoxidase genes, using molecular methods. Representative bacteria were then screened for their ability to produce tyrindoxyl sulphate in a potassium bromide supplemented media, using liquid chromatography–mass spectrometry (LC-MS).

## 2. Results

### 2.1. Bacterial Isolation

Thirty-two distinct bacteria were isolated in total (Table 1). Six, ten and sixteen bacterial morphotypes were cultured from the hypobranchial gland homogenates, hypobranchial gland dorsal swabs, and hypobranchial gland ventral swabs, respectively (Table 1). All of the distinct types of bacteria were recovered on marine agar at 7.2 pH, and no additional bacteria were cultured by incorporating the hypobranchial gland aqueous extract into the media. Despite the fact that the pH of the hypobranchial gland has been found to be highly acidic [41], no bacterial colonies were observed at pH 4.5 on marine agar, with or without a 10% aqueous gland extract, when incubated at 25 °C. A subset of the bacteria were recorded on the other growth media—25% of isolates grew on cetrimide agar (CA), 72% on blood agar (BA), 78% on marine agar with hypobranchial gland extracts (MAH) and 81% on thiosulphate citrate bile salts sucrose agar (TCBS) (Table 1).

All 32 bacterial isolates were motile, with 29 identified as Gram negative and three as Gram positive (*Bacillus* sp., *Bacillus thuringiensis* and *Halobacillus* sp.) (Table 1).

### 2.2. Molecular Identification of Cultivated Bacteria

Analysis of 16S rRNA gene sequences of the 32 cultured bacteria revealed 26 *Vibrio* spp., two *Bacillus* spp., one *Phaeobacter* sp., one *Shewanella* sp., one *Halobacillus* sp., and one *Pseudoalteromonas* sp. (Table 1).

Four *Vibrio* spp. (KR338857, KR338858, KR338859, KR338870) and two *Bacillus* spp. (KR338869, KR855712) were identified from the hypobranchial gland homogenates (Table 1); seven Vibrio spp. (KR338845, KR338846, KR338851, KR338853, KR338854, KR338856, KR338871), one *Phaeobacter* sp. (KR338852), one *Shewanella* sp. (KR338855) and one *Pseudoalteromonas* sp. (KR338872) were identified from hypobranchial gland dorsal swabs (Table 1); 15 *Vibrio* spp. (KR338844, KR338847-KR338850, KR338860-KR338863, KR338865-KR338868, KR338873, KR338874) and one *Halobacillus* sp. (KR338864) were identified from the hypobranchial gland ventral swabs sample (Table 1).

The sequence similarity to partial 16S rRNA gene sequences, available in the NCBI GenBank for isolates, ranged from 97% to 100% (Table 1).

### 2.3. Putative Bromoperoxidase Gene Screening by PCR

PCR of DNA derived from the 32 distinct bacterial isolates, using primer pairs BBFp (CCCATG TGG ACC ACC CTT TAT) and BBRp (TAA GTG GTC GAT CTT GGGAAT), amplified putative bromoperoxidase coding gene sequences from two *Bacillus* spp., but failed to amplify any DNA from the remaining 30 bacterial isolates. BLASTN comparison of the gene sequences amplified from the two *Bacillus* spp., against the NCBI database, revealed a 97% sequence similarity with *Bacillus thuringiensis* MC28- bromoperoxidase (CP003687.1) (Table 2).

### 2.4. Bacterial Extract Analysis for Brominated Compounds by Liquid Chromatography–Mass Spectrometry

Pure cultures of the bromoperoxidase containing *Bacillus* sp. and *Bacillus thuringiensis* and a subset of bromoperoxidase negative bacterial species (*Vibrio chagasii*, *Pseudoalteromonas* sp. and *Phaeobacter* sp.) were analysed for the possible production of brominated compounds using LC–MS. Evidence for the presence of tyrindoxyl sulphate was found in extracts from the two *Bacillus* spp., but not in the other three bacteria. An HPLC peak with a retention time of around 14 min was found in cell pellet extracts from *Bacillus* sp. (KR338869) and *Bacillus thuringiensis* (KR855712) cultures (Figure 1B,C). This peak corresponded to a peak detected using selected ion monitoring (SIM) at *m/z* 224, 226, [M–H]− for 6 bromoisatin (C_8_H_2_BrNO_2_), which is a stable rearrangement ion commonly detected in Tyrian purple precursors produced by *D. orbita* [67]. The peak detected using SIM at *m/z* 224, 226 was not detected in the broth control (Figure 1D), but corresponded to the peak in a purified tyrindoxyl sulphate standard (Figure 1A) isolated from the hypobranchial gland, and confirmed by proton nuclear magnetic resonance ^1^H NMR (600 MHz, CD_s_CN, 25^o^C aromatic protons δ 7.65 (1H, d), 7.55 (1H, d), 7.20, 1H, dd), methyl protons δ 2.5 (3H, s)) [67]. Tyrindoxyl sulphate did not produce a strong signal in the total ion current–mass spectrum (TIC-MS) in the negative or positive ion modes (Figure 1 and Figure 2, respectively). Nevertheless, consistent with previous characterisations of tyrindoxyl sulphate from *D. orbita* hypobranchial gland extracts [67], major ions in the negative ion mode obtained at the apex of this peak, were *m/z* 336, 338 (Figure 1 A, B, C), which corresponds to the molecular ion [M–H]− of tyrindoxyl sulphate (C_9_H_7_Br^79^NO_4_S_2_^-^, C_9_H_7_Br^81^NO_4_S_2_^-^). The diode array revealed UV absorption maxima at 228 and 302 nm from the tyrindoxyl sulphate standard and *Bacillus* spp. extracts (Figure 1A–C).

Using dianion resin extracts from the culture supernatant, we were again able to identify a peak corresponding to tyrindoxyl sulphate, at around 14 min, in a hypobranchial gland extract from *D. orbita* (Figure 2A), *Bacillus* sp. (Figure 2B) and *Bacillus thuringiensis* (Figure 2B). In the positive ion mode, the [M2H]+ ion (*m/z* 338, 340) was detected, along with a number of other paired ions, characteristic of brominated compounds Br^79^, Br^81^ (Appendix A). No peaks corresponding to tyrindoxyl sulphate, or any other brominated compounds were detected in extracts from the supernatant of *Pseudoalteromonas* sp. (KR338872), *Phaeobacter* sp. (KR338852), or *Vibrio chagasii* (KR338845) cultures (Figure 2 D–F), or the broth control (Figure 2G).

We further analysed additional supernatant dianion extracts from the two *Bacillus* against the tyrindoxyl sulphate standard in the negative ion mode with SIM 224, 226. Despite eluting slightly later, at 15 min (Appendix A), the relevant peaks were detected with molecular ion, confirmed at *m/z* 336, 338, although in the *Bacillus thuringiensis* supernatant, the peak was below the limit of detection for UV–Vis and, therefore, below the limit for accurate quantification. Quantification of tyrindoxyl sulphate in the broth extracts of *Bacillus* sp. was undertaken, using the procedure outlined by Valles-Regino et al. [67], and was estimated to be 1 mg/10 mL.

## 3. Discussion

This study provides the first evidence of bromoperoxidase producing bacteria that are capable of biosynthesizing the brominated precursor of Tyrian purple in the hypobranchial gland of a muricid mollusc. Tyrian purple is a dye of historical importance that traditionally could only be obtained by extraction from the Muricidae. Only 1g of dye is obtained from approximately 1,200 snails [31], highlighting the need for sustainable production methods, if the natural Tyrian purple dye is to be supplied on an industrial scale. Using small-scale culture without optimised conditions, we were able to obtain an estimated 1 mg of the precursor tyrindoxyl sulphate from 10 mL of *Bacillus* culture. Although 6,6′-dibromoindigo can be chemically synthesized [68,69,70] there is still a demand for the natural product. Targeting natural shellfish populations to supply the dye can place populations at risk—as demonstrated by the decline of the central American Muricidae *Plicopurpura pansa* populations—due to overharvesting [71]. Presently, *P. pansa* is considered a threatened species and is under special protection from the Mexican government [71]. However, there has been a renewed interest in natural shellfish dyes, from the Jewish community [16]. Bacteria that are capable of brominating indoxyl sulphate to generate Tyrian purple precursors, provide a potential alternative for sustainable production of this natural dye, if the culture and production can be scaled up in the future.

The low microbial diversity observed in the hypobranchial gland homogenates and the identification of 25 *Vibrio* spp. is consistent with previous studies [40,41]. However, this study also identified two *Bacillus* species which have coding sequences for bromoperoxidase enzymes. Bacterial species belonging to the *Bacillaceae* family are known to produce bromoperoxidase [72], along with several other bacteria [45,46,73,74]. Bromoperoxidases produced by marine bacteria are often involved in the biosynthesis of halogenated natural products of pharmacological importance [75], and this enzyme has the capability of reacting with indole, specifically in the 6′ position, for the production of 6-brominated indoles [75,76]. Bromoperoxidase activity has been previously reported in the hypobranchial glands of *D. orbita* [37] and other muricids [35]. *Bacillus* sp. have also been previously detected in the hypobranchial glands of *D. orbita*, using culture-independent bacterial profiling [40]. This study confirms that a bromoperoxidase associated with *Bacillus* in the hypobranchial gland of a Muricidae, is capable of brominating an indole precursor in the 6′ position, on the aromatic ring, to form tyrindoxyl sulphate. Bacterial biosynthesis of tyrindoxyl sulphate provides opportunities for sustainable production of the anti-cancer and anti-inflammatory indole derivatives from Muricidae molluscs [13,25].

Other types of bacteria isolated from the hypobranchial gland, did not contain a bromoperoxidase gene, including *Pseudoalteromonas* sp., *Phaeobacter* sp. and *Vibrio chagasii*, and as expected, these failed to produce brominated indoxyl sulphate precursors in culture. Marine *Pseudoaltermonas* have previously been found to contain halogenase enzymes and produce small polyaromatic brominated secondary metabolites [77,78]. Given that no brominated compounds were detected from the *Pseudoaltermonas* cultures in this study, this bacterium is less likely to play a role in Tyrian purple precursor synthesis. On the other hand, *Bacillus* sp. have been isolated from the egg masses of *Concholepas concholepas* [79], another Muricidae species that produces Tyrian purple in its hypobranchial glands and egg masses [11]. The fact that these bacteria are associated with the egg masses indicates possible maternal transmission of the bacterial symbionts. Overall, this study identifies the potential association between Muricidae and *Bacillus*, for bioactive-brominated indole and Tyrian purple precursor production. This could be further tested by screening other Muricidae species for bromoperoxidase containing *Bacillus* species.

The majority of other bacteria cultured from the hypobranchial glands were identified as Vibrionaceae. The finding that *Vibrio* spp. are the dominant bacterial species in the hypobranchial glands of *D. orbita* is consistent with our previous metagenomic study on *D. orbita* hypobranchial glands [40]. Our previous culture study showed that the Vibrios isolated from *D. orbita* are capable of producing indole [41]. Indeed a range of *Vibrio* spp. isolated from marine invertebrates and fish, are known to produce indoles [47,48,80,81,82], but to date, there are no reports of bromoperoxidase genes being isolated from marine *Vibrio* spp. Consistent with this, none of the *Vibrio* spp. isolated in our study contained coding sequences for putative bromoperoxidases and tyrindoxyl sulphate was not detected in culture extracts from a representative *Vibrio* sp. (*V. chagasii*). Nevertheless, it is possible that the high abundance of *Vibrio* spp. in the hypobranchial gland of *D. orbita* contribute indoxyl sulphate precursors, which are then brominated by *Bacillus* sp. for Tyrian purple precursor synthesis. This would involve a novel interaction between distinct endosymbiotic bacteria in the hypobranchial glands of Muricidae, which requires further investigation. Furthermore, a recent transcriptome study on *D. orbita* identified a tryptophanase gene, which can convert tryptophan to indole, as well as a number of genes involved in sulphur metabolism, indicating that the mollusc itself might have the capacity to produce indoxyl sulphate. Therefore, it is possible that at least part of the biogenic pathway for Tyrian purple precursors exists in the molluscs and other symbiotic bacteria, thus, potentially contributing to a mixed biosynthetic origin.

Overall, this study provides evidence that *Bacillus* spp. containing bromoperoxidase enzymes occur in the hypobranchial gland of *D. orbita*, and are capable of producing brominated precursors for Tyrian purple biosynthesis. However, there remains a possible role for marine *Vibrio* spp. in contributing non-brominated indoles, which provide the scaffold for bromination and generation of the ultimate precursor tyrindoxyl sulphate. Hence, the role of symbiotic bacteria in the biosynthesis of Tyrian purple precursors is highlighted and provides a scope for future studies on potential sustainable production of this natural dye and other bioactive 6-bromoindole derivatives, through the application of bacterial culture or genetic engineering.

## 4. Materials and Methods

### 4.1. Sample Collection, Preparation and Culturing

*D. orbita* (*n* = 15 snails) were collected from subtidal and intertidal rocky reefs near Ballina (28°84′ S and 153°60′ E), Northern NSW, Australia. Samples were collected during low tide on 11 December 2014 under permit number F89/1171-6.0 issued by Primary Industries, NSW Government, Australia. Snails were transferred live to the Southern Cross University and processed immediately. The hard shells were removed and the snails dissected, according to Westley and Benkendorff [10]. Hypobranchial glands (Figure 3) were removed aseptically, under laminar flow, and rinsed three times with sterile seawater to remove any external bacteria loosely associated with the gland. An aqueous extract of the hypobranchial gland was prepared separately for incorporation into bacterial culture media, by homogenising 2 g hypobranchial gland (15 snails) with 35 mL of phosphate buffer saline (PBS) solution in a blender. The extract solution was filter sterilized through a 0.25 μm syringe filter (Minisart, Sartorius, Sigma-Aldrich, Castle Hill, NSW Australia), before adding to the autoclaved marine agar. The pH of the hypobranchial gland was measured, using a pH microprobe (Orion, pH Micro Electrode, Thermo Scientific, Brisbane, Qld, Australia) and was found to have a mean of 4.5 (± 0.08 st. dev, *n* = 3).

The culture of potential *D. orbita* hypobranchial gland microbial symbionts was undertaken using five different growth media—marine agar (pH 7.2), marine agar (pH 4.5), marine agar and hypobranchial gland extract (pH 7.2), marine agar and hypobranchial gland extract (pH 4.5), blood agar, TCBS (thiosulphate citrate bile salts sucrose) agar and cetrimide agar. These media were chosen on the basis of their potential to provide favourable conditions, which might not be provided by the standard growth media. Marine agar with hypobranchial gland extract was used to mimic the natural environment of the *D. orbita* hypobranchial gland. TCBS and cetrimide agar was used as a selective media for isolating *Vibrio* sp. and *Pseudomonas* sp., respectively [83,84,85]. Blood agar was used as an enriched media to isolate fastidious bacterial symbionts [86]. Marine agar and marine agar supplemented with 10% aqueous gland extract plates were used at pH 7.2 and adjusted to pH 4.5, using small amounts of HCl, in order to match the pH of the hypobranchial gland lumen.

Three approaches were used to isolate and culture bacteria from the hypobranchial glands. The sampling approaches included: (1) sampling homogenates of whole hypobranchial glands; (2) taking dorsal swabs of glands and (3) taking ventral swabs of glands. In all cases, three hypobranchial glands, with an approximate total weight of 0.25 g, were used. Homogenates were prepared using a sterile mortar and pestle, whereas, swab samples were taken using sterile cotton swabs. Each of the samples was diluted in 9 mL of sterile sea water, mixed thoroughly by vortexing, and three-fold dilutions were prepared with sterile seawater. Additional concentrated homogenates were also prepared from six individual hypobranchial glands. Following Marinho et al. [87], these were homogenised, separately, in marine broth at a concentration of 1 g/mL, then directly plated onto marine agar and cetrimide agar, for maximum recovery of the bacterial symbionts, including *Pseudomonas* sp.

A 100 μL aliquot from each homogenate and swab sample was spread onto the duplicate agar plates. The agar plates were incubated for 14 days at 25 °C. Agar plates were observed daily for bacterial colonies and the colony size and morphology were recorded. Morphologically distinct colonies were selected for Gram-staining, using standard procedures [41], and molecular identification of the isolates. All genetically distinct isolates were screened for bromoperoxidase genes and a subset of these were analysed for brominated indole production.

### 4.2. 16S rRNA Sequencing of Bacterial Isolates

Morphologically distinct colonies were subjected to 16S rRNA sequence analysis. DNA was extracted using Qiagen DNA extraction kits (QIAmp DNA mini kit, Qiagen, Chadstone, Vic, Australia). PCR reactions comprised 2.5 µL of 10× PCR buffer; 2.5 µL of dNTPs (2mM), 1.25 µL of 50 mM MgCl_2_; 1 µL genomic DNA (35–80 ng); 0.4 µL *Taq* polymerase and 1 μL forward primer (27F) (10 μM), 1 μL reverse primer (1492R) (10 μM), 15.35 μL Milli-Q water in a final volume of 25 µL. PCR cycle conditions consisted of an initial denaturation at 94 °C, for 5 min, followed by 30 cycles of 45 s at 95 °C, 1 min at 58 °C and 1 min at 72 °C. PCR amplicons were separated by agarose gel electrophoresis and visualised by GelRed staining under UV irradiation. Positive PCR products were purified using the QIAquick PCR Purification Kit (Qiagen), in accordance with the manufacturer’s instructions, and sequenced by Macrogen Inc, (Seoul, Korea). DNA sequences were analysed using the sequence scanner software v1.0 and compared with sequences in the NCBI GenBank database, by BLASTN. All 16S rRNA gene sequences from the 32 bacterial isolates were lodged with GenBank, under the accession number KR855712, KR338844-KR338874. The isolates were preserved by diluting 1:1 in sterile marine broth containing 30% glycerol and then stored in −80 °C for further analysis. Gram staining (100× magnification, Olympus, Macquarie Park, NSW Australia) was performed on the pure cultures.

### 4.3. Bromoperoxidase Gene Screening

The genomic DNA of the 32 bacterial isolates were screened for the bromoperoxidase genes, using primer pair BBFp (CCCATG TGG ACC CTT TAT) and BBRp (TAA GTG GTC GAT CTT GGGAAT). These primers were designed on the basis of bromoperoxidase consensus sequence, derived from five *Bacillus* strains [88]. The PCR reaction was composed of 2.5 µL of 10× PCR buffer; 4 µL of dNTPs (2mM), 1 µL of 50 mM MgCl_2_, 2 µL genomic DNA, 0.5 µL *Taq* polymerase, 1.5 μL forward primer (FP) (10 μM), 1.5 μL reverse primer (RP) (10 μM), 12 μL Milli-Q water in a final volume of 25 µL. PCR cycle conditions comprised an initial denaturation at 94 °C, for 5 min, followed by 30 cycles of 1 min at 94 °C, 1.30 min at 58 °C and 1 min at 72 °C. PCR amplicons were separated and visualised by agarose gel electrophoresis and GelRed staining, under UV irradiation. Positive bromoperoxidase fragments, approximately 700 bp in size, were purified and sequenced by Applied Biosystems 3730 and 3730xl capillary sequencers (Australian Genome Research Facility, Brisbane, Australia), were further analysed using the sequence scanner software v1.0 (ThermoFisher, Brisbane, Australia) and the sequences were compared with the NCBI GenBank database by BLASTN. Nucleotide sequences for putative bromoperoxidase genes were submitted to the NCBI GenBank under accession number KT180165 and KT180166.

### 4.4. Liquid Chromatography–Mass Spectrometry (LC–MS) Analysis of Bacterial Extracts

Extracts from the pure subcultures of *Bacillus* sp., *Bacillus thuringiensis*, *Vibrio chagasii*, *Pseudoalteromonas* sp. and *Phaeobacter* sp. were analysed for the possible production of any brominated compounds. Bacteria were grown in sterilized Schott bottles containing 60 mL marine broth and tryptone broth, since these contain tryptophan which is suspected to be the ultimate precursor for Tyrian purple [23], and were supplemented with 0.2 g potassium bromide (KBr). The bacterial growth was maintained for 24 h at 25 °C. A control of tryptone broth without bacterial inoculation was also maintained with KBr, for comparison. Exponentially growing cultures were centrifuged at 6000 rpm (Heraeus Instruments Biofuge pico, ThermoFisher, Brisbane, Australia) for 10 min, to separate the cells and the supernatant. Extraction of the supernatant was performed by ion exchange chromatography, by passing the supernatant through a dianion resin (Dianion HP 20, Supelco, Bellefonte, PA, USA), then washing the column with methanol, before drying the methanol extract using a stream of high purity nitrogen gas. The cell pellet was also extracted in chloroform—methanol (1:1) and dried under nitrogen gas. These extracts were analysed using LC–MS.

LC–MS analysis was undertaken using an Agilent 1260 infinity (Santa Clara, CA, USA) High Performance Liquid Chromatography (HPLC) system, coupled with a 6120 Quad mass spectrometer (MS), according to the procedure described by Valles-Regino et al. [67]. Tyrindoxyl sulphate in the bacterial extracts was quantified against a standard calibration curve, using HPLC, by integrating the area under the curve for absorbance at 210 nm [67]. The HPLC utilized a Phenomenex luna C18 reversed phase column (100 × 4.6 mm), with a solvent gradient from 10% to 95% acetonitrile (ACN) with 0.005% trifluoroacetic acid (TFA) over 18 min, at a flow rate of 0.75 mL/min. Peak absorption was monitored using parallel UV–Vis diode-array detection (DAD). Electrospray ionisation (ESI) mass spectrometry was used in the positive and negative ion modes. Selected ion monitoring at *m/z* 224, 226 was also used in negative mode to detect these common fragment ions that are typically generated in the mass spectrum, from the Tyrian purple precursors in the extracts from *D. orbita* [67]. Agilent ChemStation (Agilent Technologies Australia, Mulgrave, Vic) was used to analyze the LC–MS data. Characteristic ion cluster patterns from Br^79^ and Br^81^ in the mass spectrum were used for identifying the presence of any brominated compounds [10]. To confirm the identity of the purified tyrindoxyl sulphate standard H^1^ MNR was undertaken, and was recorded on a Bruker Advance III HD 500 MHz spectrometer (Bruker Biospin, Alexandria, NSW, Australia) in CD_3_CN and D_2_O (Novachem, Cambridge Isotopes Laboratories, Tewksbury, MA, USA). ^1^H chemical shifts were referenced to either CD_3_CN (1.96 ppm) or D_2_O (4.80 ppm).

## Figures and Tables

**Figure 1 marinedrugs-17-00264-f001:**
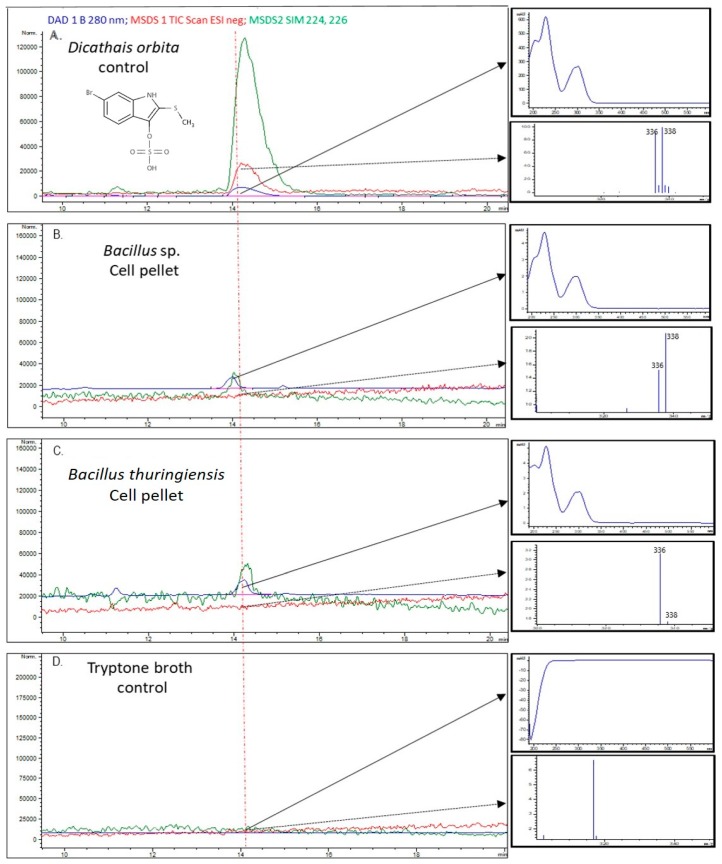
Tyrindoxyl sulphate control (**A**) and chloroform extracts from the cell pellets of two *Bacillus* species (**B**,**C**) cultured from the hypobranchial glands of *Dicathais orbita* and a corresponding tryptone broth control supplemented with KBr (**D**). Left panels show the HPLC scan at 280 nm in the diode array (blue), total ion current (TIC) in the negative ion mode (red) and selected ion monitoring for major fragment ions at *m/z* 224, 226 (green). Right panels show the UV–Vis spectra and mass spectrum obtained from the apex of the major peak obtained at 14.1–14.26 min.

**Figure 2 marinedrugs-17-00264-f002:**
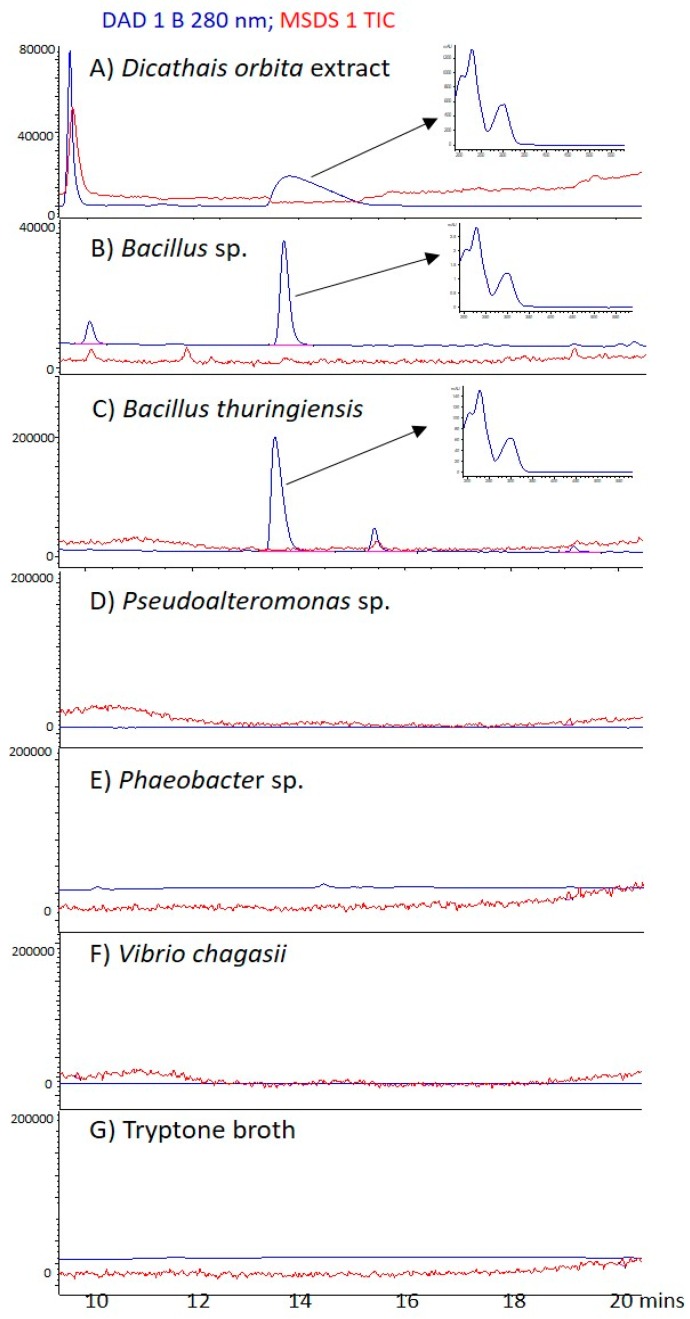
Liquid chromatography–mass spectrometry analysis of diaion resin chromatography extracts of (**A**) *Dicathais orbita* hyprobranchial glands; (**B**) *Bacillus* sp. (KR338869); (**C**) *Bacillus thuringiensis* (KR855712); (**D**) *Pseudoalteromonas* sp. (KR338872); (**E**) *Phaeobacter* sp. (KR338852); (**F**) *Vibrio chagasii* (KR338845) culture supernatant and (**G**) Tryptone broth control (supplemented with KBr). The red lines are from the total ion current in the mass spectrum in the positive ion mode. The blue lines represent the HPLC scan at 280 nm and the inset panels show the UV–Vis scan from the diode array for the major peak, eluting at 14 min.

**Figure 3 marinedrugs-17-00264-f003:**
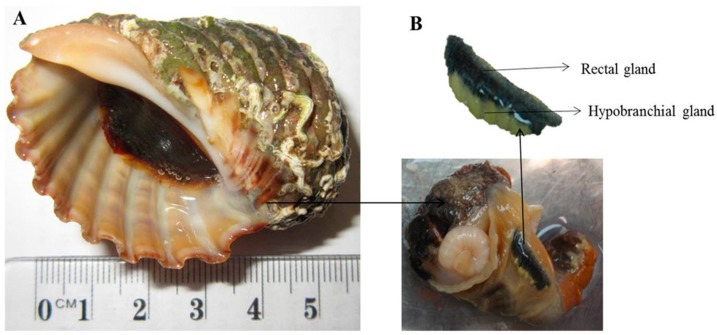
*Dicathais orbita* hypobranchial gland used for isolating and culturing bromoperoxidase and indole producing bacteria. (**A**) *Dicathais orbita*; (**B**) Hypobranchial and rectal gland.

**Table 1 marinedrugs-17-00264-t001:** Nucleotide Basic Local Alignment Search Tool (BLASTN) analysis of partial 16S rRNA gene sequence derived from bacteria isolated from *D. orbita* hypobranchial glands cultured on different agar media. Species identification is based on the closest match to NCBI GenBank data.

Closest Match and Accession Number	Agar Media ^1^					GenBank Accession Number	Length (Base Pair)	Identity (%)	Gram Stain	Hypobranchial Gland Preparation ^2^
	MA *	TCBS	CA	MAH	BA					
*Vibrio* sp. (KM369853.1)	+	+	-	-	+	KR338844	1149	100	Gram-	HGVS
*Vibrio chagasii* (NR117891.1)	+	+	+	+	+	KR338845	1083	99	Gram-	HGDS
*Vibrio alginolyticus* (KF886646.1)	+	+	+	+	+	KR338846	1115	99	Gram-	HGDS
*Vibrio* sp. (KP126921.1)	+	+	-	-	+	KR338847	510	100	Gram-	HGVS
*Aliivibrio* sp. (FR744854.1)	+	+	-	-	+	KR338848	1087	99	Gram-	HGVS
*Vibrio azureus* (JF412237.1)	+	+	-	-	+	KR338849	1119	99	Gram-	HGVS
*Vibrio pomeroyi* (KM014017.1)	+	+	-	-	+	KR338850	1090	99	Gram-	HGVS
*Vibrio* sp. (KM369851.1)	+	+	+	+	-	KR338851	1004	100	Gram-	HGDS
*Phaeobacter* sp. (GQ906799.1)	+	+	-	+	-	KR338852	1044	100	Gram-	HGDS
*Vibrio* sp. (GQ406789.1)	+	+	+	+	-	KR338853	1046	99	Gram-	HGDS
*Vibrio* sp. NB0059, (KP770076.1)	+	+	+	+	+	KR338854	1096	100	Gram-	HGDS
*Shewanella* sp. (JF825445.1)	+	+	+	+	+	KR338855	1058	98	Gram-	HGDS
*Vibrio mediterranei*, (HF541948.1)	+	+	-	+	+	KR338856	1116	99	Gram-	HGDS
*Vibrio* sp. (KF188532.1)	+	-	-	+	-	KR338857	1096	100	Gram-	HGH
*Vibrio* sp. (KF188531.1)	+	-	-	+	-	KR338858	1066	99	Gram-	HGH
*Vibrio* sp. (HG942391.1)	+	-	-	+	-	KR338859	626	99	Gram-	HGH
*Vibrio* sp. (KP126921.1)	+	+	-	+	+	KR338860	951	97	Gram-	HGVS
*Vibrio* sp. (KM369860.1)	+	+	-	+	+	KR338861	796	99	Gram-	HGVS
*Vibrio* sp. (KM369853.1)	+	+	-	+	+	KR338862	1091	100	Gram-	HGVS
*Vibrio* sp. (GQ406789.1)	+	+	-	+	+	KR338863	1079	100	Gram-	HGVS
*Halobacillus* sp. (FM992846.1)	+	+	-	+	+	KR338864	1109	100	Gram +	HGVS
*Vibrio* sp. (FJ457587.1)	+	+	-	+	+	KR338865	1004	100	Gram-	HGVS
*Vibrio* sp. (KM369853.1)	+	+	-	+	+	KR338866	1038	99	Gram-	HGVS
*Vibrio splendidus* (AB038030.1)	+	+	-	+	+	KR338867	1083	100	Gram-	HGVS
*Vibrio harveyi* (KR003734.1)	+	+	-	+	+	KR338868	973	100	Gram-	HGVS
*Bacillus* sp. (KJ756140.1)	+	-	-	+	-	KR338869	1034	100	Gram +	HGH
*Bacillus thuringiensis* (KC355253.1)	+	-	-	+	-	KR855712	1180	99	Gram +	HGH
*Vibrio* sp. (HF937138.1)	+	-	-	+	-	KR338870	866	99	Gram-	HGH
*Vibrio* sp. (EU340847.1)	+	+	+	+	+	KR338871	1110	99	Gram-	HGDS
*Pseudoalteromonas* sp. (KP301110.1)	+	+	+	+	+	KR338872	973	99	Gram-	HGDS
*Vibrio* sp. (FJ457361.1)	+	+	-	-	+	KR338873	1056	99	Gram-	HGVS
*Vibrio jasicida* (AB562594.1)	+	+	-	-	+	KR338874	907	99	Gram-	HGVS

* indicates that the bacterial isolates cultured on MA were subcultured for screening the bromoperoxidase enzyme; ^1^ MA—marine agar; TCBS—thiosulphate citrate bile salts sucrose; CA—cetrimide agar; MAH—marine agar with hypobranchial gland extracts; BA—blood agar; + indicates isolated; - indicates not isolated on the relevant growth media; ^2^ HGVS, Hypobranchial gland ventral swabs; HGDS—Hypobranchial gland dorsal swabs and; HGH—Hypobranchial gland homogenates.

**Table 2 marinedrugs-17-00264-t002:** BLASTN analysis showing the closest match in the NCBI GenBank for the putative bromoperoxidase gene in two *Bacillus* spp.

Bacterial Isolates	GenBank Accession Number	Length (Base Pair)	Identity (%)	Closest Match, Accession Number, Position and Protein ID
*Bacillus* sp.	KR338869	628	97	*Bacillus thuringiensis* MC28- bromoperoxidase (CP003687.1); 2239980-2240816 and AFU13721.1
*Bacillus thuringiensis*	KR855712	634	97	*Bacillus thuringiensis* MC28- bromoperoxidase (CP003687.1); 2239980-2240816 and AFU13721.1

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
