# Peer review of "Bromoperoxidase Producing Bacillus spp. Isolated from the Hypobranchial Glands of A Muricid Mollusc Are Capable of Tyrian Purple Precursor Biogenesis"

_marinedrugs, 2019, doi:10.3390/md17050264_

Round 1
Reviewer 1 Report
The paper has been significantly improved during the review process and should now be accepted for publication.
Author Response
We thank the review for their kind support
Reviewer 2 Report
This manuscript describes the isolation of a number of bacteria from the mollusc D. orbita. Of which, two Bacillus sp. have been shown to code sequences for bromoperoxidase enzyme and produce tyrindoxyl sulphate. The manuscript is difficult to read/follow, is very short and sometimes confusing in results sections. It can be only considered for acception upon a number of major improvements, particularly in the description of the spectrometric data in results section. A very serious proof reading is also necessary as there are many typos, grammatical mistakes or format type of mistakes. I summarize the most important revisions needed from my point of view below:.
Title:
-Please change to: Bromoperoxidase producing Bacillus ‘sp.’ (or spp) ….
-LCMS is a hyphenated technique and should be shown as LC-MS in the abstract and in the whole paper
Introduction:
-Line 66.. Replace ‘ are obtained’ with ‘derive’ or ‘originate’
-L93.-94. Should read as: A … study , or, Recent ... studies. Check grammatically
Results:
-This section should be probably renamed as Results and Discussion and there is no Discussion heading?
-Section 2.1. Please add the abbreviation used for media. Also add 1-2 lines which bacteria were isolated from which part of the mollusk, which is highly important for the study (See & consider moving lines 118-124 from section 2.2. to section 2.1 which is most relevant place). A table 1 is given but no proper summary has been included on the table.
Section 2.2. The title of 2..2 includes ‘indole producing bacteria’ but no information or results are given here. This subsection and in general the results section is very short.
-There should be a section describing the results of microbial culture (any color formation etc) and the extraction for example extraction yield, which is crucial. Also crucial is the yield of tyrindoxyl sulphate for different bacteria.
2.3. Clearly mention if the other isolated bacteria are positive/negative for production of bromoperoxidase enzyme (alos later on , or indole or tyrindoxyl sulphate) as such statements come later in discussion and become confusing. And elaborate this section, which is again too short. It gives a table but that is mostly it.
Section 2.4. Which is the most problematic section.
Line 143. This first sentence is not understandable and start with a missing microorganism name (instead a genBank accees number). Therefore I ma not clear if other microbes isolated in the study contain indole or tyrindoxyl sulphate.
-Line 146. What are m/z 224 and 226 values are for and for which compound? If these are characteristic fragments for tyrindoxyl sulphate then mention it here. The reader is left alone to interpret or search what these are. Also are these [M+H]+ or [M-H]- peaks? This info lacks everywhere in the whole manuscript, but should be added
-Convert all m/z to Italics.
-Line 146. Figure D should be Figure 1D?
-where does the used standard tyrindoxyl sulphate derive for. Give reference if relevant.
-Line 147. tyrindoxyl sulphate (not Tyrindoxyl ..)
-Line 148. The name of the technique is ‘proton’ nuclear magnetic resonance and the abbreviation is 1H NMR, not H1 NMR (or MNR as shows up later Line 358, section 4.4.). Also correct this sentence, it is not proper English.
-You need a reference for the 1H NMR data given in lines 148-149 to prove your statement there . As aforementioned rephrase the sentence that it is clear what these data are proving?
Line 154. Give tyrindoxyl sulphates molecular formula containing both Bromine isotopes. E.g. CxHyOzNx79Br and CxHyOzNx81Br.
-If more than one Figures are being referred to, please mention it as (Figures 1 and 2)
-Line 156. Add ‘we’ after supernatant?
Line 159. Is M2H+ correct and correctly written ?
-Line 162. Why only Vibrio cahagasii (remove the dott after it) was mentioned for being tyrindoxyl sulphate negative although 26 Vibrio sp were isolated?
-The Latin name of bacillus thuringiensis is written in all kind of versions in the text (e.g. line 158), Tables and Figures.
Line 164. What is SIM 224,226. m/z, [M+H]+ or [M-H]- ions??
Figure 1 legend. Line 173. Correct the type: supplmentated
Line186. … bromoperoxidase producing bacteria that ‘are’ capable. In the same sentence the authors should also discuss briefly the yield of the pigment from the microbial broth as they mention this issue in lines 189-190 from mollusk and it is highly relevant to compare such quantitative facts with the results of their own study.
Lines 200-201. There was no mention of indole positive bacteria (Vibrio) where does this come from? See my previous comment above.
L210. Bacillus should be Italics
Line 223. The fact that these bacteria ‘are’
-Discussions starting from line 228 need the results on the presence/absence of indole and/or tyrindoxyl sulphate in Vibrio species, to be able to follow the discussions.
Line 242. Remove double dots.
Materials & Methods.
-No subsection on general procedures, reagents etc?
-Section 4.1. Line 271. Do authors mean homogenate instead of extract here: hypobranchial gland extract (pH 7.2), marine agar and hypobranchial gland extract. I understand something different from extract, prepared with a solvent.
Section 4.4. Should include Extraction in its title and the yields etc of the extracts.
Line 354. Add m/z before 224 and 226. Move their description to results section.
Line 361. Is ACN-d6 correct?
References. Problems with ref. 27, 39, 41, 46, 60, 61, 70, 73, 78.
Author Response
Reviewer 2
Please see attached file.

Round 2
Reviewer 2 Report
The manuscript has been improved but there are still many small typos to be corrected , e.g. the NMR solvent used in page 6, ionisation in MS data should be superscript, m/z values are not all in Italics, multiple Figures are written singular etc. The authors should do a better proof reading later on and correct these scientifically important typos.
This manuscript is a resubmission of an earlier submission. The following is a list of the peer review reports and author responses from that submission.
Round 1
Reviewer 1’s comments and authors’ responses:
The manuscript by Ngangbam et al. provides evidence for a bacterial biosynthesis of Tyrian purple in Mollusc associated basteria that have been identified as Bacillus strains. The paper is sound and well written.
Nevertheless, the authors go a bit far when interpreting their results. For example, in the Abstract we find the statement „this study provides evidence that symbiotic Bacullus sp. could be the ultimate source of Tyrian purple“. This may turn out to be true but is at the moment not justifed based on the data presented.
Response: We agree that we do not have conclusive evidence that Bacillus is the “ultimate source of Tyrian purple”. We have reworded the title to:
“Bromoperoxidase producing Bacillus isolated from the hypobranchial glands of a muricid mollusc are capable of Tyrian purple precursor biogenesis”.
We have also reworded the concluding statement in the abstract (lines 34-36):
“This study supports a potential role for symbiotic Bacillus sp. in the biosynthesis of Tyrian purple”.
It is also possible that biogenetic pathways for the production of the pigments exist in bacteria and in the molluscs as has been shown for numerous other examples where identical molecules can be found e.g. in plants and in endophytic fungi.
Response: We believe it is most likely a mixed biosynthesis in this case. We have undertaken extensive genomic studies on the hypobranchial glands of Dicathais orbita and have never been able to identify a bromoperoxidase or halogenase enzyme that indicates a capacity to add a bromine group on the 6 position of the indole ring, as is required for the Tyrian purple precursor. These studies include a comprehensive transcriptome published in Marine Drugs (Baten et al 2016, Mar. Drugs 14, 135 doi:10.3390/md14070135) which encompassed 216,218,545 high quality reads with 219,437 contigs and a total assembly length of 117,767,308 base pairs. In this previous paper we specifically looked for enzymes potentially involved in Tyrian purple precursor synthesis and we identified aryl sulfatase, tryptophanase and enzymes involved in sulfur reduction and transfer but despite undertaking a comprehensive BLAST search against all known bromoperoxidase genes, we found no matches. We have subsequently undertaken further sequencing of the hyrpobranchial glands and have examined differential gene expression against the foot tissue and again found no bromoperoxidase or halogenase genes (unpublished data). We also previously published a suppressive subtractive hybridization study on genes upregulated in the hypobranchial glands (Laffy et al 2013 Comparative Biochemistry and Physiology Part D 8:111-122) and again found aryl sulfatase but no bromoperoxidase genes. Further unpublished work in Patrick Laffys PhD thesis (Flinders University) used primers designed form a range of known bromoperoxidase genes but was unable to isolate and matching sequences from the mollusc tissue.
Nevertheless, we do agree that some genes involved in Tyrian Purple precursor synthesis are produced by the mollusc. This was explicitly mentioned in the discussion (lines 240-241):
“….indicating the mollusc itself may have capacity to produce indoxyl sulfate. ”
We have now also revised the last sentence in this paragraph referring to a possible mixed biosynthesis origin line (268-270):
“Therefore it is possible that at least part of the biogenic pathway for Tyrian purple precursors exists in the molluscs and their symbiotic bacteria, thus potentially contributing to a mixed biosynthetic origin.”
Likewise, in the Discussion the authors state "the discovery that murid bacterial symbionts contribute to Tyrian purple precursors biosynthesis provides an alternative and more sustainable method to produce this dye“ goes too far. We dont know anything about the quantities of the pigments produced by the bacteria or whether it may be possible to scale the process up in the future. This knowledge is essential for discussing the production of the pigments by bacteria.
Response: The point we intended to make was that bacteria provide an option for biosynthesis that is likely to be more sustainable than wild harvest of the molluscs, given that Muricid populations are already know to crash from over-harvesting. However, we agree that more work is required to ensure scale- up is feasible. Therefore, we have reworded this statement in the discussion (lines 221-224) as:
“Bacteria that are capable of brominating indoxyl sulfate to generate Tyrian purple precursors provide a potential alternative for sustainable production of this natural dye, if the culture and production can be scaled up in the future.”
Reviewer 2’s comments and authors’ responses:
Ngamgbam et al. report isolation of 32 bacteria from the muricid Dicathais orbita. Among these, two Bacillus sp. have been –putatively - found to contain bromoperoxidase genes by PCR. The same bacteria have been shown by preliminary LC-MS methods to contain the monobromo indole derivative tyrindoxyl sulphate, the ‘brominated’ precursor of the Tyrian purple. Hence the authors conclude these Bacillus sp to be the ‘ultimate’ biosynthetic origin of Tyranian purple.
This overall aim of the study is interesting however the work suffers from many shortcomings especially from chemistry point of view, briefly listed below. Hence it represents only an early, preliminary study, and this reviewer cannot recommend its publication in this journal.
Response: We disagree with the suggestion that this is a preliminary study. The results are novel and conclusive in terms of the isolation and identification of bacteria that produce bromoperoxidase enzymes from the hypobranchial glands of a Muricid mollusc and evidence that those bacteria can produce tyrindoxyl sulfate. I have been working with tyrindoxyl sulfate and Tyrian purple precursors from Muricidae mollusc for over 20 years now and our lab has developed a very specific method for the identification of this compound by LCMS (Valles-Regino et al 2016. Molecules, 21, 1672 doi:10.3390/molecules21121672 ). I have absolutely no doubt the compound identified as Tyrindoxyl sulfate in the extracts of Bacillus sp. was in fact this compound based on retention time in the LC, UV profile and the mass spectral fragmentation pattern (see revised Figure 1 & 2, page 8 & 10 and Supplementary Figures S1 & S2). The extracts also produce a compound corresponding to tyrindoxyl sulfate that turn purple in sunlight by TLC (data not shown but for method see Nongmaithem et al 2017. Scientific Reports 7 article number 1704 https://www.nature.com/articles/s41598-017-17551-3#Sec16 ).
The reviewers criticism of the “short-comings” from a “chemistry point of view” appear to be based on the fact that the Bacillus culture was indole negative in the biochemical test - this has been addressed in detail below.
1. This study at most is a very preliminary indication that Tyrian purple may have bacterial origin. Similar dyes have been reported from plants or bacteria, hence it could be somehow expected. The main problem is however that many other isolated bacteria (e.g. Vibrio sp.) are indole positive while these 2 Bacillii, which are supposed to be producing the precursor, are indole negative. To my opinion, this maybe a very strong argument against their statement and require further in-depth studies prior to publication. How can an enzyme work if the indole structure to be brominated is missing ? But then the authors identify the tyrindoxyl sulphate in extracts? There is probably a basic chemistry problem. Furthermore, in discussions the authors speculate that the co-isolated Vibrio sp. may provide the indoles to Bacillii. This should have been studied first and if positive, included into this publication.
Response: Firstly we need to clarify that we Tyrian purple itself has never been reported from a bacteria and at present it is only known to be produced by Muricidae molluscs and perhaps a marine worm, from the precursor tyrindoxyl sulfate after hydrolysis by an aryl sulfatase enzyme produced by the marine invertebrate (or artificially by acid). Therefore, we are not suggesting that Tyrian purple has a purely bacterial origin, although it is true that indigo (the non-brominated equivalent nature dye) is produced by bacteria and plants. Nonetheless, this does not detract from the novelty of isolating a bacteria that contains a bromoperoxidase gene from the hypobranchial glands of the Tyrian purple producing molluscs and demonstrating that this bacteria is capable of producing a brominated indoxyl sulfate precursor of Tyrian purple in culture.
It was surprising that the Bacillus sp. did not produce indole positive reaction using Kovac’s reagent and we thank the reviewer for raising this interesting point for further consideration. There are many reasons why we may have a negative or false negative result in this basic screening assay. It is possible that the Bacillus don’t produce a simple indole ring that reacts with p-dimethylaminobenzaldehdye to produce a red soluble product. Rather, they may be capable of generating indole sulfate directly and could form the brominated product, which may not react and/or produce the same red pigment on reaction with Kovacs reagent. Other compounds produced by the Bacillus may suppress the formation of the acid end products with Kovac’s reagent or it is possible that the Bacillus require bromine ions to initiate a reaction from tryptophan to generate the brominated indoxyl sulfate compound.
In light of the fact that Kovac’s reagent is typically used to assist in the identification of bacteria and is less suitable for the chemical identification of indole products we now feel that this tests adds little to that paper and in fact could be misleading. Therefore we have now removed all reference to the indole test, including results lines 130-132, Table 1 indole column and Discussion lines 253-255. We have revised the discussion test lines 255-256 to:
“The majority of other bacteria cultured from the hypobranchial glands were identified as Vibrionaceae”
In general Bacillus and other bacterial species identified from D. orbita previously are known to produce bromoperoxidase enzyme (but also indoles). Therefore novelty effect to contain these enzymes is also somehow low. Important is that the enzyme and indole are identified in the same biological material and more sophisticated work proves the idea.
Whilst bromoperoxidase enzymes have previously been reported from bacteria in the Bacillaceae family, this is the first study to 1) isolate Bacillus from the hypobranchial glands of a mollusc; 2) demonstrate that the Bacillus sp. cultured from the hypobranchial glands contain bromoperoxidase genes and 3) demonstrate that these isolates have the capacity to produce tyrindoxyl sulfate in culture. These are major novel advances that contribute to our understanding of the bacterial communities in a unique gastropod biosynthetic gland and potential biogenic pathways for Tyrian purple precursor synthesis. This work is significant considering the historical and current use of Tyrian purple and the fact that it can only be obtained in natural form from Muricidae molluscs at present. Although further work will be required to upscale for sustainable production, we do provide evidence that the Bacillus not only produces the bromoperoxidase enzyme, but also that it can produce tyrindoxyl sulfate, a brominated indoxyl sulfate in culture. The indole negative result and its limitations are discussed above, but the detection of indole is far less important than demonstrating the Bacillus has the ability to produce the actual brominated indoxyl sulfate precursor molecule. There is never conclusive proof in science but our results provide reasonable support for the hypothesis that symbiotic Bacillus sp. could contribute to the production of Tyrian purple in Muricidae.
2. Another important point is the authors feed the culture medium with KBr themselves artificially. This is probably the origin of the bromine atom in tyrindoxyl sulphate. Have they done cultures without KBr for comparison? This is to me also another problem. They have done tryptone broth without bacterial inoculation but not without KBr?
Response: The media was supplemented with KBr deliberately to provide a source of bromine ions that the bacteria could use to brominate indoxyl sulfate. The relevant negative control including tryptone broth with KBr but without bacterial inoculation is shown in Figure 1 D and Figure 2 G, as well as supplementary Figure S2. I have now revised the Figure legend to explicitly explain the KBr supplementation in the tryptone broth controls (lines 191 and 204). These broth controls all confirm that tyrindoxyl sulfate does not form without the bacteria. This control is more important that a tryptone broth without KBr and with bacterial inoculation, which would only establish whether or not the Bacillus could produce tyrindoxyl sulfate without extra bromine ions. Future studies could optimise the specific conditions required for maximum production of tyrindoxyl sulfate, but this was outside of our study.
We also confirm that tyrindoxyl sufate was not produced in KBr tryptone broth supplemented culture of several other bacteria that do not contain the bromoperoxidase enzyme (Figure 2 D, E and F). Tyrindoxyl sulfate was only produced in two Bacillus cultures and was detected in two independant the supernatant extracts (Figure 2 B and C, Supplementary Figure S1 & S2), as well as the cell pellets (Figure 1 B & C) providing strong evidence that the isolated bacillus can produce the brominated indoxyl sulfate Tyrindoxyl sulfate in culture, in the presence of bromine ions.
We have undertake elemental analyses on Dicathais orbita using inductively coupled plasma mass spectrometry and found high levels of bromine in the tissues of this molluscs (unpublished data). It is not clear how the mollusc accumulates the bromine at this stage but that is outside the scope of the current study. Nevertheless, it does justify the supplementation of the bacterial cultures with KBr in this study and it would be unrealistic to expect the Bacillus to synthesize detectable quantities of tyrindoxyl sulfate in the absence of bromine.
3. Several other points need to be mention include:
- Tyrian purple is a dibromo indole. How do the author imagine this reaction is happening from the monobromo precursor? Another enzyme?
Response: The production of 6,6’ dibromoindigo (Tyrian purple) from the monomer Tyrindoxyl sulfate is well established in the literature, including the two cited reviews by Cooksey (ref 32 and 32 cited line 58) and in reference 23, the Marine Drugs review by Benkendorff, 2013 (Marine Drugs, 11, 1370-1398), which was cited in the paper (but not correctly in the endnote list - now updated lines 53 and 61). To clarify the formation of Tyrian purple form the precursor monomer I have added the following sentence lines 64-66.
“The brominated indoxyl sulfate precursors are hydrolysed by an aryl sulfatase enzyme, which is produced by the mollusc [38] and then spontaneously reacts with oxygen, dimerises and is photolytically cleaved to form the final dye pigment [23,39].
- This reviewer missed to find information how he authors searched indole specifically in the extracts??
We were not looking for indole in the extracts but rather for the brominated indoxyl sulfate Tyrian purple. This was done using LC- MS analysis as described in the methods section 4.4 lines 376-393, with the results presented in section 2.4 lines 151 - 187 and Figure 1 & 2.
The work has been largely based on the detection of bromoperoxidase enzyme in 2 Bacilli, but it is only 1 sentence is given in the Introduction about its importance for this work. This is extremely slim considering its importance.
This study involved four main components 1) the isolation of bacteria from the hypobranchial glands under different culture conditions, 2) the genetic identification of the isolated bacteria, 3) screening for bromoperoxidase genes and 4) the detection of tyrindoxyl sulfate production in extracts form the cultured bacteria. Therefore, all of these aspects of the paper are covered in the Introduction section. Bromoperoxidase is mentioned three times in the introduction: 1) The role of bromoperoxidase enzymes in tyrindoxyl sulfate precursor synthesis is mentioned in lines 62-64; 2) the ability of bacteria to produce bromoperoxidase enzymes is mentioned in lines 74-76 and 3) the aim of screening isolates for bromoperoxidase genes is mentioned in line 99.
- The authors use very unusual culture media but never explain why they use them in the results (or any other appropriate) section.
The reasons for the selected culture media are explained in the methods section 4.1 lines (303- 310) as follows:
“These media were chosen based on their potential to provide favourable conditions which may not be provided by standard growth media. Marine agar with hypobranchial gland extract was used to mimic the natural environment of the D. orbita hypobranchial gland. TCBS and cetrimide agar was used as a selective media for isolating Vibrio sp. and Pseudomonas sp. respectively [86-88]. Blood agar was used as enriched media to isolate fastidious bacterial symbionts [89]. Marine agar and marine agar supplemented with 10% aqueous gland extract plates were used at pH 7.2 and adjusted to pH 4.5 using small amounts of HCl in order to match the pH of the hypobranchial gland lumen.”
- The results are slim but also elaborated very little with a few sentences maximum. It is unclear if all culture extracts of 2 Bacilli contained tyrindoxyl sulphate?
The results contain four sections in total originally encompassing lines 107 - 169 plus 2 Tables and 2 Figures legends. In the revised paper we have included two supplementary Figures and more details in Figure 1 as well as the corresponding text (lines 146-207). We believe this is adequate for describing the results given that all interpretation of the results in included in the discussion (lines 212-280).
All extracts taken from the 2 Bacilli were found to contain some evidence for the presence of Tyrindoxyl sulfate in the MS, although on one occasion the UV-Vis was below the limit of detection. The LCMS results for tyrindoxyl sulfate detection in the cell pellets of both Bacillus species are now expanded in lines 151-169. The supernatant dianion resin extracts are explained in line 174 - 187.
- Table 1. The abbreviations used for tissue sources should be explained, as they have not been explained anywhere else.
The abbreviations for the tissue source in Table 1 have been defined in the Table footnote immediately below the Table (after the media explanations).
- The LC-MS analysis of the whole extract results should be included in the paper to let the reader to check.
I am not exactly sure what the reviewer means by this, but in the interests of clarity in the results presentation, we have focused on the region where tyrindoxyl sulfate is known to occur based on the standard. Full spectra from 0-30min do not add any value because the peaks < 10mins are mainly associated with the solvent front and after 30min with the column wash. They do not contain any additional brominated or indole compounds. Whilst providing full chromatograms is possible, it provides a messier trace at lower resolution for no clear benefit (see examples below).
We have however, now included the selected Total Ion Current and Selected Ion Monitoring in the figures and added new figures to illustrate the results for cell pellets and the supernatants, with the later in positive and negative ion mode (Supplementary Figure 1 & 2).
- Figure2. The chemical structure of the tyrindoxyl sulphate need to be corrected. The NH should be in the ring system.
We apologize for this oversight and have now corrected the structure.
- Where is the Figure 1a-1i?? There is a Fig.1A-D but this is only the LC-MS chromatograms of the extracts.
In Figure 1 the “i” is used to indicate the Tyrindoxyl sulfate peak in parts D, E and F. However we have now replaced this figure and use arrows to indicate the signals for the relevant peaks in the chromatograms.
- Many problemes with references
o Some references have full jour nalnames, some abbreviated.
o The open access journals lack doi numbers.
o Ref. 67, journal name is missing.
o Some references are unclear if these are books, or journals, or something else (e.g. 16)
o Please check all references for whole format and correctness
The references have been checked and corrected as required. We used the MPDI endnote library format - which has some problems that revert whenever the reference list is updated. This format also doesn’t seem to include the DOI in the references - so we have manually added these where available.
Reference 67 Journal title added
Reference 16 is a book and the publisher and place of publication is provided in the correct format